# Effect of Ginger on Chemical Composition, Physical and Sensory Characteristics of Chicken Soup

**DOI:** 10.3390/foods10071456

**Published:** 2021-06-23

**Authors:** Wen Duan, Li Liang, Yan Huang, Yuyu Zhang, Baoguo Sun, Lina Li

**Affiliations:** 1Beijing Key Laboratory of Flavor Chemistry, Beijing Technology and Business University, Beijing 100048, China; 15754367187@163.com (W.D.); gcfll@126.com (L.L.); huangyan_916@163.com (Y.H.); sunbg@btbu.edu.cn (B.S.); 2Jinmailang Food Co., Ltd., Xingtai 054000, China; linda18833993893@163.com

**Keywords:** ginger, chicken soup, taste components, sensory characteristics, correlation analysis

## Abstract

In order to investigate the effect of ginger on taste components and sensory characteristics in chicken soup, the content of amino acids, organic acids, 5′-nucleotides, and mineral elements were determined in chicken soup sample. With the ginger added, free amino acids in chicken soup obviously increased and exceeded the total amounts in ginger soup and chicken soup. The content of glutamic acid (122.74 μg/mL) was the highest among 17 free amino acids in ginger chicken soup. Meanwhile, six organic acids detected in chicken soup all obviously increased, among which lactic acid (1523.58 μg/mL) and critic acid (4692.41 μg/mL) exceeded 1000 μg/mL. The content of 5′-nucleotides had no obvious difference between ginger chicken soup and chicken soup. Compared with chicken soup, ginger chicken soup had a smaller particle size (136.43 nm) and color difference (79.69), but a higher viscosity. With ginger added in chicken soup, the content of seven mineral elements was reduced, and the content of total sugar increased. Results from an electronic tongue indicated a difference in taste profiles among the soups. The taste components and sensory quality of chicken soup were obviously affected by adding the ginger.

## 1. Introduction

Chicken is popular for its delicious taste, low fat, high protein and rich in various essential fatty acids. Compared with beef, pork, and mutton, chicken is more delicate and richer in nutrients and flavor components. At present, the main cooking methods of chicken include steaming, stewing, and boiling. Among them, stewing is more widely used for the soup product. Chicken soup is regarded as a good tonic with therapeutic function [1]. Micro-/nanocolloids in soup have important biological activities, which can change the absorption and utilization behavior of nutrients and bioactive compounds [2]. The colloidal characteristics of those nanoparticles have been preliminarily investigated. This research provided a novel perspective to understanding the complexity of soups [3]. Boiling soup can release the nutritional components and flavor substances, such as high-quality protein, creatinine, functional peptides, flavor peptides, flavor nucleotides, and free amino acids, in chicken, making chicken soup delicious and easy to digest and absorb [4,5].

The flavor is often the key factor affecting consumers’ acceptance and preference of food, particularly for soups. Therefore, several studies published in the last two decades have focused on the identification and characterization of the flavor-imparting components of soups. Zhan et al. (2020) analyzed the relationship between nonvolatile flavor compounds in chicken soup and sensory evaluation, and they found that aspartic acid, glutamic acid, glycine, alanine, and proline strongly affected the umami taste in chicken soup [6]. Kong et al. (2017) isolated, purified, and identified flavor compounds in chicken soup and chicken enzymatic hydrolysates. They found that the content of flavor compounds in the enzymatic hydrolysate was higher than that in chicken soup [7]. In addition to flavor compounds, sensory characteristics such as color, zeta potential, and viscosity could also be used to measure the overall taste of soup. Zou, Xu, Zou, and Yang (2021) compared the differences in color, particle size, zeta potential, viscosity, flavor, mineral content, purine, and total triglyceride content of soups made by different processing methods to understand the effect of dietary soup on health. The flavor characteristics of cooked soup were obviously better than those of steamed soup, which proved the importance of cooking technology on soup [8]. Some studies have shown that the release of nutrients in soup is not only attributed to the occurrence of the Maillard reaction, but also related to the nanostructured components in soup [9].

Ginger is a kind of seasoning that is often used in soups and other meat products to remove fishy odor and greasiness and improve its flavor [10,11]. Muhialdin, Kadum, Fathallah, and Hussin (2020) found that adding fermented ginger sauce to chicken products could promote the growth of microorganisms and enhance their antibacterial ability. As a conventional spice, ginger has the advantage of enhancing flavor [12]. According to the traditional cooking method of stewing chicken, adding ginger increases the flavor of chicken soup. The effect of ginger on the flavor of chicken soup is of great significance for the industrial production of chicken soup. At present, there are few studies on the effect of ginger on the flavor of chicken soup, so it is urgent to further explore.

The aims of this work were to investigate the effects of ginger on the chemical composition and physical and sensory characteristics of chicken soup. By comparing the color, particle size, potential, viscosity, total sugar, amino acids, organic acids, nucleotides, and mineral elements of three samples, ginger soup, chicken soup, and ginger chicken soup, the influence of ginger on the actual taste of chicken soup could be fully determined. The relationship between chemical composition and physical and sensory quality was also analyzed.

## 2. Materials and Methods

### 2.1. Materials and Chemicals

Sanhuang chicken was purchased in Yonghui market (Beijing, China) with the content of moisture 73.00 g/100 g, ash 3.00 g/100 g, crude protein 17.00 g/100 g, crude fat 8.00 g/100 g, and total sugar 0.50 g/100 g. These values in Sanhuang chicken were determined according to the national standard method. The ash, crude protein, crude fat, moisture, and total sugar were according to GB 5009.3-2016 (the first method), GB 5009.5-2016 (the first method), GB/T 5009.6-2016 (the second method), GB 5009.3-2016, and GB/T 9695.31-2008 (the second method), respectively. Fresh ginger was purchased from the Yonghui market (Beijing, China). The salt was purchased in Zhongyan Yangtze Salinization Co., Ltd. Oxalic acid, lactic acid, citric acid, succinic acid, L-(+)-ascorbic acid, pyroglutamic acid, and potassium dihydrogen phosphate dodecahydrate phosphate (all AR grade) were obtained from Sinopharm Chemical Reagent Co. (Shanghai, China). Potassium dihydrogen phosphate (KH_2_PO_4_), phosphoric acid (H_3_PO_4_), hydrochloric acid (HCl), and disodium hydrogen phosphate dodecahydrate (Na_2_HPO_4_•12H_2_O) (all AR grade) were purchased from Sinopharm Chemical Reagent Company (Shanghai, China). Inosine 5′-monophosphate (5′-IMP), adenosine 5′-monophosphate (5′-AMP), guanosine 5′-monophosphate (5′-GMP), and cytidine 5′-monophosphate (5′-CMP) were purchased from Sigma-Aldrich (St. Louis, MO, USA). Durashell AA analytical reagents, including an internal standard solution, was purchased from Tianjin Bona Agel Technology Co., Ltd. Methanol and acetonitrile (ACN) (HPLC grade) were purchased from Fisher Scientific (Shanghai, China). Ultrapure water was purchased from Hangzhou Wahaha Group Co., Ltd. (Hangzhou, China). Sulfosalicylic acid (AR grade) was obtained from Biochemical Technology Co., Ltd. (Shanghai, China).

### 2.2. Preparation of Soup

Ginger soup: Rinsed and drained ginger was mixed with water (ginger–water = 1:100 (*w*/*v*)) and stewed in a nutrient soup model with a stewing temperature of 100 °C for 3 h, cooled to room temperature, then filtered with a No. 4 qualitative filter paper (Ge Biotechnology (Hangzhou) Co., Ltd., Hangzhou, China), centrifuged at 4 °C, 10,000 r/min for 10 min to remove impurities, and packed into a tube for standby.

Chicken soup: According to the method of Fan et al. (2018) [13]. All Sanhuang chickens were ca. 1000 g in weight and were stored at −20 °C in polyethylene bags until use. The head, claw, neck, and visible fat were removed according to naked eye observation. Then, the rest was blanched with boiling water for 1 min and stewed with 2 times weight of water in a nutrient soup model with a stewing temperature of 100 °C for 3 h. After the chicken soup was cooled to room temperature, it was filtered with a No. 4 qualitative filter paper (Ge Biotechnology (Hangzhou) Co., Ltd.), centrifuged at 10,000 r/min for 10 min to remove impurities, and packed into a tube for standby. For each batch of chicken, three replicates were performed, while every two samples of soups of the same batch of chicken were combined, resulting in three samples of soups for the subsequent analysis.

Ginger chicken soup: The head, claw, neck, and visible fat were removed from Sanhuang chicken. The rest was blanched with boiling water for 1 min and added to 2 times weight of water. Two grams of ginger (100 g meat) were added, and the mixture was stewed in a nutrient soup model with a stewing temperature of 100 °C for 3 h, cooled to room temperature, then filtered with a No. 4 qualitative filter paper (Ge Biotechnology (Hangzhou) Co., Ltd.), centrifuged at 10,000 r/min for 10 min to remove impurities, and packed into a tube for standby.

### 2.3. Physical Characterization of Soup Properties

#### 2.3.1. Zeta Potential and Particle Size Estimation

According to the method of Li et al. (2020a) [14], a zeta potential analyzer (Zetasizer Nano ZS-90, Malvern Instruments, Worcestershire, UK) was used to measure the zeta potential and particle size of three samples. The sample solution (1 mL) was added to the sample pool, and then the electrode was immediately inserted into the battery to start the test. The influence of impurities should have been avoided. Each sample was paralleled three times. The test conditions were as follows: temperature: 20 °C; dispersant name: water; dispersant refractive index: 1.330; viscosity: 1.00 cp. The RI and viscosity of water at 20 °C were 1.333 and 1.00 cp, respectively. The particle size data was analyzed by Malvern particle size analysis software.

#### 2.3.2. Rheological Behavior and Dynamic Oscillatory Rheological Measurements

Rheological measurement was carried out by the method of Zhu, Bhandari, and Prakash (2020a) with some modifications [15]. The modular rotation and interface rheometer Discovery HR-1 (TA Instruments, Model DHR-1, Elstree, UK) was used to measure, and the diameter of the parallel plate clamp was 40 mm. Parameter setting: plate clamp clearance was 1000 μm; soaking time was 30 s; gap was 1000; temperature was raised from 25 to 37 °C; rotation speed was 0.1 to 100 r/s. Sample was balanced for 2 min before measuring, redundant sample was wiped off on the edge of sensor, and the relationship among shear rate, viscosity, and shear stress was analyzed.

#### 2.3.3. Color Parameters Analysis

The L*, a*, and b* values of three samples were determined by KONICA MINOLTA chroma meter CR-400 [16]. Each sample was measured at least three times in parallel, where the L* value represents the brightness of the sample (L* = 0 is black, L* = 100 is white), the corresponding value range from dark to bright was 0~100, and the higher the value is, the higher the brightness is and the whiter the color of the soup is; a* represents the red and green degree of the sample (“+” represents the red degree, “−” represents the green degree); and b* represents the yellow and blue degree of the sample (“+” represents the yellow degree, “−” represents the blue degree).

### 2.4. Total Sugar Analysis

According to the sulfuric acid phenol method [17], slightly modified. The standard curve of glucose was drawn: 0, 1, 2, 3, 4, and 5 mL of glucose standard solution was put into 50 mL volumetric flask, obtaining concentrations of 0, 20, 40, 60, 80, and 100 μg/mL, respectively. One milliliter of the above glucose standard solution was absorbed, transferred to a colorimetric tube, and mixed with phenol solution. Then, 5 mL of concentrated sulfuric acid was added to the mixture, placed at room temperature for 20 min, and measured for its absorbance value at 470 nm wavelength. A standard curve was drawn with glucose content as abscissa and absorbance value as ordinate. The 1 mL sample was added in a 5 mL volumetric flask; 1 mL of solution was taken from the volumetric flask, put into a small beaker, and mixed with 1 mL phenol solution, then added to 5 mL concentrated sulfuric acid, shaken up immediately, and placed at room temperature for 20 min. At the wavelength of 470 nm, with anhydrous glucose as the standard, its absorbance value was measured and the content of polysaccharide in the sample was calculated.

### 2.5. Free Amino Acids Analysis

According to a previously described method [18], we weighed 1 mL soup, added sulfosalicylic acid (10%) in the ratio of 1:1, stood for a period of time, centrifuged (9600 r/min, 10 min, 4 °C) the sample, took the supernatant through a 0.45 μm filter membrane, and waited for the sample to be analyzed by HPLC (Agilent Technology Co., Ltd., Beijing China).

The free amino acids in the samples were analyzed by Durashell AA kit. Agilent 1260 high performance liquid chromatography (HPLC) was used to analyze 17 kinds of free amino acids in the standard and sample. The 17 kinds of free amino acids were Asp, Glu, Ser, Pro, Gly, Thr, Ala, Val, Met, Ile, Phe, Lys, Leu, Arg, His, Tyr, and Cys-Cys. Determination conditions: the chromatographic column was Durashell AA (4.6 mm × 150 mm, 3 μm); the mobile phase A was (9.50 g sodium tetraborate decahydrate and 9.00 g disodium hydrogen phosphate dodecahydrate were weighed, respectively; 2000 mL ultrapure water was added; pH was adjusted to 8.20 with 36% hydrochloric acid); the mobile phase B was (45% acetonitrile, 45% methanol, and 10% ultrapure water); the gradient elution method was 6–10% B 0–6 min, 10% B 6–8 min, 10–16% B 8–10 min. The flow rate of the mobile phase was 1.60 mL/min; the column temperature was 45 °C; the detection wavelengths were 338 nm and 262 nm, respectively.

### 2.6. Organic Acids Analysis

The 5 mL sample was mixed with perchloric acid (5%) in a ratio of 1:1 (*v*/*v*), held for 5 min, centrifuged (9600 r/min, 4 °C) for 10 min, and the supernatant was filtered through a 0.45 μm filter membrane for analysis by HPLC (Thermo Scientific, Waltham, WA, USA).

Chromatographic column: venusil XBP C18 (4.6 mm × 250 mm, 5 μm); column temperature: 25 °C; detector: ultraviolet absorption detector; detection wavelength: 205 nm; mobile phase: methanol–potassium dihydrogen phosphate (0.01 mol/L, pH = 2.8) = 5:95 (*v*:*v*); flow rate: 1.00 mL/min; injection volume: 10 μL; isocratic elution; determination for 3 times. Preparation and quantitative method of the standard curve: the external standard method was used for quantitative analysis, and the concentrations of the standard mixed solution of organic acid were 4.00, 1.00, 0.50, 0.10, 0.05, and 0.01 mg/mL. The organic acid standard solutions with different concentration gradients were injected under the same conditions, and the peak area of the obtained chromatogram was plotted against the concentration to draw the standard curve of the organic acid [19].

### 2.7. 5′-Nucleotides Analysis

The sample preparation was consistent with that of Section 2.6 (Organic Acids Analysis).

Chromatographic column: venusil XBP C18 (4.6 mm × 250 mm, 5 μm); column temperature: 25 °C; detector: ultraviolet absorption detector; detection wavelength: 254 nm; mobile phase: methanol–potassium dihydrogen phosphate (0.05 mol/L) = 5:95 (*v*:*v*); flow rate: 1.00 mL/min; injection volume: 10 μL; isocratic elution; determination for 3 times. Preparation and quantitative method of standard curve: external standard method was used for quantitative analysis. The standard mixed solutions of 5′-CMP, 5′-GMP, 5′-IMP, and 5′-AMP were prepared with concentrations of 2.00, 0.50, 0.10, 0.05, 0.01, and 0.005 μg/mL. The standard solution of taste nucleotides with different concentration gradients was injected under the same conditions. The peak area of the chromatogram was plotted against the concentration, and the standard curves of four nucleotides were drawn [20].

### 2.8. Equivalent Umami Concentration

The equivalent umami concentration (EUC, g MSG/100 g) refers to the idea that the taste produced by 100 g sample can be equal to the mass of sodium glutamate (MSG) [21]. The calculation formula refers to literature [20].

### 2.9. Mineral Analysis

The mineral content in soups were measured as described in the literature with some modifications [8]. The 5 mL sample was accurately transferred into the digestion tank, 6 mL of concentrated HNO_3_ was added, and the samples were predigested at 120 °C without cover for 30 min until no yellow smoke bubbles were generated, then taken out and cooled to room temperature of 25 °C. The two-stage digestion method was used in the microwave digestion instrument. The first stage was 150 °C for 10 min; the second stage was 180 °C for 15 min. After microwave digestion, the acid was removed at 160 °C to about 1 mL of the remaining sample and the constant volume was taken out to 25 mL. An ICP-9800 (Shimadzu, Japan) was used to determine metal elements. Instrument parameters: high frequency power: 1.20 kW; plasma gas: 10.00 L/min; auxiliary gas: 0.60 L/min; carrier gas: 0.70 L/min. The multielement standard solution was used for qualitative and quantitative analysis of mineral elements in samples by the standard curve method, and each sample was determined 4 times.

### 2.10. Electronic Tongue

According to the method of Zhao, Wei, Gong, Xu, and Xin (2020), an SA-402B (Insent) (Intelligent Sensor Technology, Inc., Kanagawa, Japan) was used to measure the taste of three samples [22]. The samples were deoiled and deslagged, then the volume was fixed to 100 mL, and then the samples were put into the special measuring cup for the electronic tongue. Before analysis, the six sensors were pretreated in the reference solution (30 mM KCl solution containing 0.3 mM tartaric acid) for 24 h, including the sensors for sourness (CA0), bitterness (C00), astringency (AE1), umami (AAE), saltiness (CT0), and sweetness (GL1). The sweetness was determined separately from the other five tastes. Before the test, the electronic tongue needed to go through self-test, calibration, and other operations. The program was set as sample taste collection time of 30 s, aftertaste collection time of 30 s, and cleaning time of 300 s. The test was conducted at room temperature of 25 °C. The test was repeated five times, and the last three response values were taken as valid data.

### 2.11. Statistical Analysis

Microsoft Excel 2010 was used for data processing; all the significant difference (*p* < 0.05) (Duncan test) and correlation analyses were performed using SPSS 22.0 (SPSS Inc., Chicago, IL, USA). All the tests were repeated 3 times, and the test data were in the form of mean ± standard deviation. Correlation analysis was performed using Origin Pro 2021.

## 3. Results and Discussion

### 3.1. Color Analysis, Particle Size, and Zeta Potential

The optical properties of food contribute to its overall sensory properties and consumer appeal [23]. The colors of three samples are shown in Appendix A. The a* and b* represent the degrees of red and yellow, respectively Appendix A. The b* value of all soups was greater than the a* value. When the a* value and b* value were positive, the larger the a* value, the redder the color, and the larger the b* value, the yellower the color. The higher the L* value, the brighter the color. With added ginger, the L* value of chicken soup decreased, and the solution turned yellow. This may be due to the decrease of particle size, which makes the soup smoother while reducing the intensity of scattered light, thus reducing the brightness. Three kinds of soups were determined by the zeta potential analyzer. The results were shown in Table 1. The average particle size of ginger soup was 969.87 nm. Compared with chicken soup, ginger chicken soup had a smaller particle size (136.43 nm). This indicates that ginger could promote the migration of more taste components from the chicken to the soup. The zeta potential of soups is summarized in Table 1. The zeta potential showed that both chicken soups were negatively charged. The higher the zeta potential magnitude (positive or negative), the higher the stability of the solution. The zeta potential of ginger soup was −20.80 mV. With added ginger, the zeta potential of chicken soup obviously decreased. A small zeta potential reflects lower stability for ginger chicken soup. The main reason was that ginger chicken soup was a complex system that contained a variety of proteins, polysaccharides, nucleic acids, and other substances [24,25]. The zeta potential affected the repulsive force between particles: higher zeta potential meant that the distance between adjacent suspended particles in the sample solution was larger and the electrostatic repulsive force was larger [26,27].

### 3.2. Viscosity Analysis

Because of the important relationship between food rheological properties and processing technology, the study of food rheological properties could help us to understand the components, internal structure, and molecular morphology of food [28,29]. The viscosity of three samples were shown in Figure 1a. The results showed that with the increase of shear rate, the viscosity of chicken soup and ginger chicken soup decreased rapidly and gradually tended to be stable, which indicates that the chicken soup and ginger chicken soup showed the characteristics of shear-thinning and pseudoplasticity. Factors such as temperature, concentration, pH, etc. could affect viscosity [30,31]. Under the same shear rate, the viscosity of ginger chicken soup was higher than that of chicken soup. This may be because ginger chicken soup contained more taste compounds than chicken soup, and the interaction between protein molecules, fat particles, and other molecules led to higher viscosity. In addition, the viscosity of soup may also be greatly affected by particle size and particle interaction [32]. Smaller particles were better dispersed in the soup, resulting in higher viscosity values [33]. The decrease of particle size made the interaction between particles stronger.

The existence of yield stress in the soup means that the interaction of macromolecules and the aggregation of particles in the soup formed a network structure, and the yield stress was needed to destroy this structure [34]. Figure 1b depicts the shear stress–shear rate curves of ginger soup, chicken soup, and ginger chicken soup, i.e., the shear stress of the three kinds of soup along with the increase of shear rate. The relationship between shear stress and shear rate was nonlinear, which indicates that the three kinds of soup were non-Newtonian fluids [35].

### 3.3. Total Sugar Analysis

As one of the most important precursors for the formation of nutritional flavor substances in soups, total sugar could have an impact on the quality of soup. The content of total sugar in three kinds of soup is shown in Table 1. The content of total sugar in ginger chicken soup was 70.58 mg/mL, which was significantly higher than that in chicken soup (36.97 mg/mL). This may be because the addition of ginger promoted the dissolution of sugar into the soup; in addition, the structure of polysaccharides in ginger may have been destroyed and dissolved in the soup [36]. The total sugar content in ginger chicken soup was less than the total amount in separate ginger and chicken soup, which may owe to the chemical reaction between polyphenols and sugar in ginger resulting in a significant decrease of sugar content.

### 3.4. Free Amino Acids Analysis

Free amino acids are flavor substances that not only play an important role in the growth of organisms, but also have the effect of improving the overall taste in food. The contents of free amino acids in the three kinds of soup are shown in Table 2. Only four kinds of amino acids were detected in ginger soup, which were consistent with the free amino acids found in 20 pungent spices by Huang et al. (2021) [37], while the amino acid composition in chicken soup and ginger chicken soup were the same. Still, there were significant differences in the content of each amino acid in the two kinds of soups. The content of total free amino acids in the ginger chicken soup was the highest (813.97 μg/mL), which indicated that ginger could promote the dissolution of protein in soup [38]. Glu and Asp are umami amino acids, which could cooperate with other substances to increase the overall taste of food. With the addition of ginger, the content of Glu and Asp in chicken soup increased significantly. The content of bitter amino acids in chicken soup and ginger chicken soup was the highest, which was consistent with the research of Zhang et al. (2020) and may be caused by protein denaturation and secondary bond breaking during heating, which lead to the exposure of hydrophobic amino acids such as arginine [39].

### 3.5. Organic Acids Analysis

Organic acids and their sodium salts, such as monosodium succinate and disodium succinate, are umami and can be added to food as flavor enhancers [40]. The content of organic acids in the three kinds of soup is shown in Table 2. It can be seen from Table 2 that only two organic acids were detected in ginger soup, and the content of oxalic acid and citric acid was significantly lower than that in chicken soup and ginger chicken soup. Six organic acids were detected in chicken soup and ginger chicken soup. The content of succinic acid in the ginger chicken soup was 927.30 μg/mL, which was significantly higher than that in chicken soup, indicating that the addition of ginger may promote the release of succinic acid in chicken soup. Wang et al. (2018) studied that succinic acid and lactic acid were the main organic acids in stewed chicken soup [41]. When these two organic acids interacted with other flavor substances, they played a contribution to the overall taste of chicken soup. In this work, citric acid and lactic acid accounted for the largest proportion of organic acids in chicken soup and ginger chicken soup. The highest content of citric acid was in ginger chicken soup at 4692.41 μg/mL; the lactic acid content in ginger chicken soup was 1523.58 μg/mL. Citric acid in chicken soup was 3999.96 μg/mL, and lactic acid was 1112.98 μg/mL, which may be a result of different processing technologies.

### 3.6. Nucleotides Analysis

The results of nucleotide determination are shown in Table 2. The nucleotides in chicken soup came from chicken, which was the contributor to the flavor of chicken soup. Flavor nucleotides include 5′-IMP, 5′-GMP, 5′-CMP, and 5′-AMP. There is also a synergistic effect between nucleotides, which plays an important role in enhancing the overall flavor. Four nucleotides were detected in ginger soup, chicken soup, and ginger chicken soup. Among the detected nucleotides, 5′-CMP and 5′-GMP were notable flavor substances because of their large proportion and significant effect on the taste of chicken soup. Except for 5′-CMP, the content of other nucleotides in the ginger chicken soup was higher than that in chicken soup, but there was no significant difference. This phenomenon was probably due to the acceleration of nucleotide release through the addition of ginger, which greatly improved the overall flavor of ginger chicken soup [42]. Wang et al. (2018) explored the change trend of nucleotide content in chicken soup under different processing methods, and the results showed that the highest content of four nucleotides in chicken soup was 5′-CMP, and the lowest was 5′-IMP, which was the same as the change trend in this experiment [41]. The content of 5′-IMP (20.50 μg/mL) was the lowest, the content of 5′-CMP (80.47 μg/mL) was the highest, and there was no significant difference between chicken soup and ginger chicken soup.

### 3.7. Equivalent Umami Concentration

The EUC value refers to the concentration of monosodium glutamate and the intensity of taste produced by the synergistic action of amino acids (aspartic acid and glutamic acid) and nucleotides (5′-IMP, 5′-GMP, and 5′-AMP). The results of EUC values are shown in Appendix A. The EUC values of chicken soup and ginger chicken soup were 16.64 g MSG/100 g and 19.54 g MSG/100 g, respectively. Chiang et al. (2007) studied chicken soup, mushroom soup, pork soup, and seafood soup, and the EUC values were 30.30, 19.10, 32.00, and 14.10 g MSG/100 g, respectively, which were inconsistent with the results of this work [43]. It may be that this soup was determined after concentration, which led to its high EUC value. With ginger added, the EUC of ginger chicken soup was significantly higher than that of chicken soup. This may be due to the high content of umami amino acids and nucleotides in ginger chicken soup, which made ginger chicken soup more delicious.

### 3.8. Mineral Analysis

Four major elements (Ca, Mg, K, and Na) and three trace elements (Fe, Cu, and Cr) in three kinds of soup were determined. Table 3 shows the content of mineral elements in the three kinds of soup. With ginger added, the content of mineral elements in the chicken soup decreased significantly. This may be due to the fact that ginger inhibits the dissolution of mineral elements in chicken soup and the aggregation of some macromolecules or colloidal particles in chicken soup through connection with many metal elements, resulting in the reduction of its content [8]. Fe is an essential trace element for the human body, but it was not detected in ginger chicken soup. This may be because ginger promoted protein in chicken soup to chelate iron and inhibited the release of free iron, which led to the decrease of Fe content [44]. It was observed that boiling Silkie chicken soup had a higher content of K, Na, Ca, and Mg than steaming it, and different cooking techniques (boiling, stewing, or steaming) led to significant differences in the mineral levels [8]. K and Na could regulate acid–base balance and maintain cell osmotic pressure, which plays an important role in human health. The contents of K in ginger chicken soup and chicken soup were 818.33 mg/L and 290.67 mg/L, respectively, and the contents of Na were 1082.5 mg/L and 393.38 mg/L, respectively. The Na content in ginger chicken soup was significantly lower than that in chicken soup. This may be because the addition of ginger inhibited the release of Na content.

### 3.9. Electronic Tongue Analysis

The results of three kinds of soup determined by electronic tongue are shown in Appendix A. The taste values of the three kinds of soup had significant differences, which showed that the electronic tongue could distinguish the tastes of different soups. These results may owe to the differences in flavor precursors and meat structure [45]. The bitterness, sourness, sweetness, and astringency of the ginger soup were the most prominent. The umami, saltiness, and richness of the chicken soup and ginger chicken soup were higher. Among them, there were significant differences in bitterness, astringency, aftertaste-B, umami, richness, and saltiness between chicken soup and ginger chicken soup. Compared with ginger soup, the increase in umami, saltiness, and richness in chicken soup and ginger chicken soup may be due to the content of umami amino acids, minerals, and total sugar, which provided soups with a good taste. The umami of ginger chicken soup was lower than that of chicken soup, but its aftertaste-B value was higher than that of chicken soup, which also endowed chicken soup with better taste.

### 3.10. Correlation Analysis

A correlation analysis was carried out among sensory evaluation and EUC, total sugar, amino acids, organic acid, nucleotides, and physical properties (zeta potential, particle size, color, viscosity, and shear stress). The taste of chicken soup could be characterized by different indicators. Chicken soup is a mixed system, so its sensory properties are related to the material composition of the system, which then affected the different tastes. As shown in Figure 2, umami had positive correlation with EUC, 5′-IMP, 5′-AMP, organic acids, amino acids, zeta potential, and viscosity. Sweetness had negative correlation with organic acids, EUC, nucleotides, and amino acids. Saltiness had significantly positive correlation with Asp, Pro, zeta potential, and viscosity. EUC values had significantly positive correlation with Asp, Glu, Gly, Pro, Arg, His, 5′-IMP, citric acids, pyroglutamic acid, and zeta potential. A correlation was also shown between sensory evaluation and components. Goudoulas et al. (2017) studied the rheological properties of the mixed material system and found that the rheological dynamics of the mixed material system were different from those of the pure material system, and the rheological properties of the mixed material system were more complex [46,47]. The EUC value had negative correlation with particle size. The results of correlation analysis indicated that the sensory properties also played a key role in the system of soup.

## 4. Conclusions

In summary, the chemical components and physical characteristics of chicken soup are related to sensory characteristics. Compared with chicken soup, ginger chicken soup had smaller particle size, higher viscosity, higher total sugar, lower stability, and richer taste. With ginger added, the content of total sugar, umami free amino acids, and sweet amino acids in chicken soup increased significantly, making chicken soup sweeter and richer. These results proved the importance of ginger to the quality of chicken soup.

## Figures and Tables

**Figure 1 foods-10-01456-f001:**
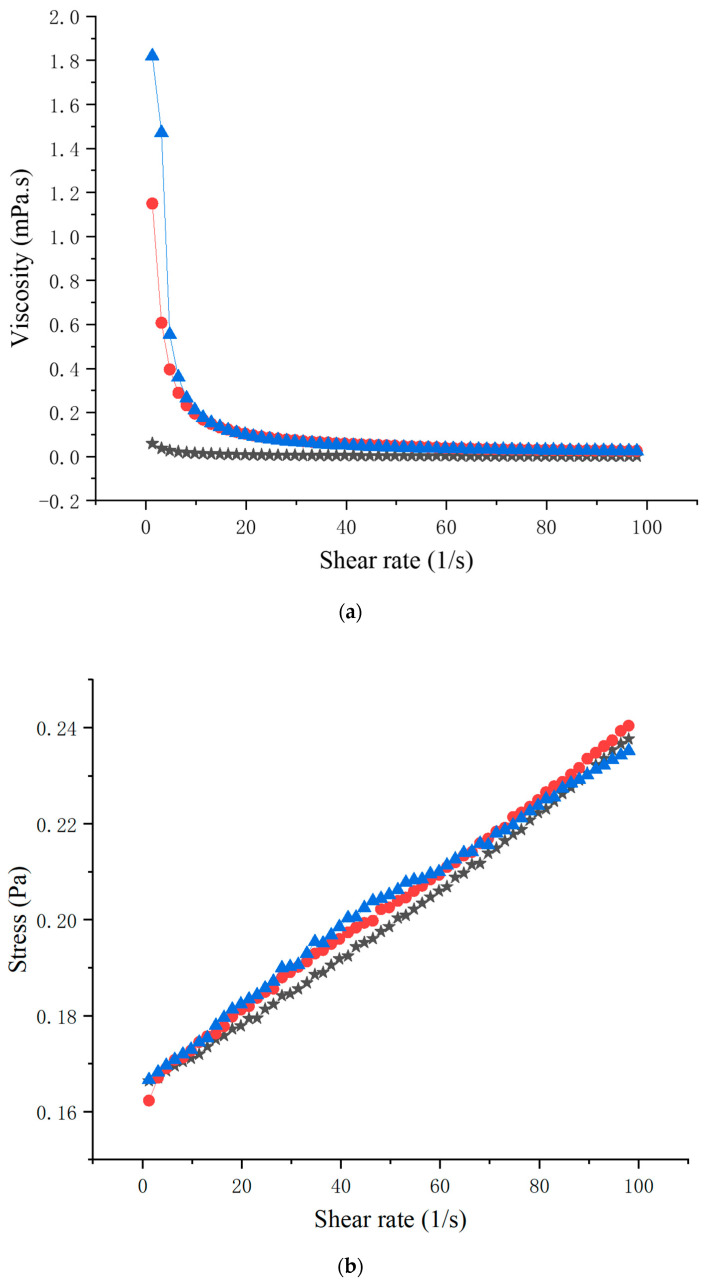
Shear rate viscosity curves of different soups (**a**); Shear stress shear rate curves of different soups (**b**). 
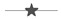
 represent ginger soup, 
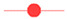
 represent chicken soup, 
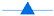
 represent ginger chicken soup.

**Figure 2 foods-10-01456-f002:**
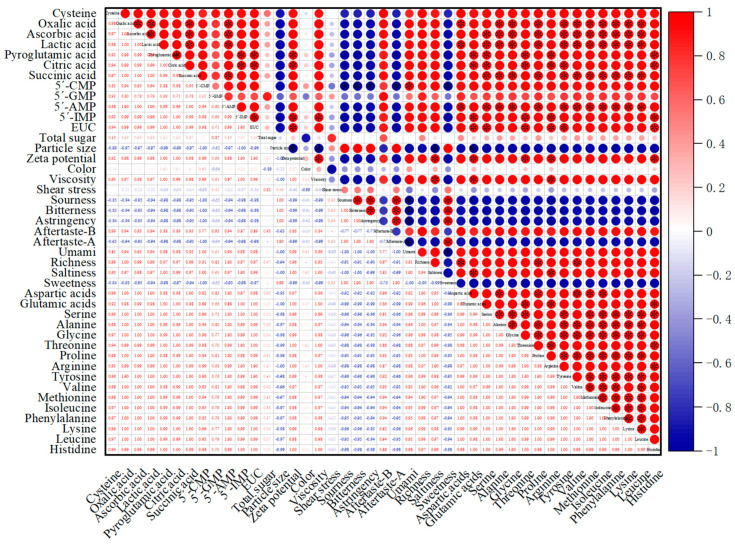
Correlation analysis of EUC, chemical properties, and sensory properties (* represents *p* < 0.05).

**Table 1 foods-10-01456-t001:** Component analysis of different soups.

Samples	Particle Size (nm)	Zeta Potential (mV)	Total Sugar (mg/mL)
Ginger soup	969.87 ± 100.02 ^b^	−20.80 ± 1.80 ^a^	49.98 ± 0.00 ^b^
Chicken soup	150.53 ± 1.91 ^a^	−6.00 ± 1.28 ^b^	36.97 ± 0.25 ^a^
Ginger chicken soup	136.43 ± 1.80 ^a^	−4.70 ± 1.16 ^b^	70.58 ± 2.66 ^c^

Means in the same row with no common superscript differed significantly (*p* < 0.05).

**Table 2 foods-10-01456-t002:** The content of free amino acids, 5′-nucleotides, and organic acids in different soups.

Compounds		Contents (μg/mL)		
		Ginger Soup	Chicken Soup	Ginger Chicken Soup
Free amino acids				
Umami	Asp (Aspartic acid)	3.86 ± 1.35 ^a^	34.88 ± 2.47 ^b^	46.23 ± 0.23 ^c^
	Glu (Glutamic acid)	n.d.	112.90 ± 0.36 ^a^	122.74 ± 0.35 ^b^
	Total	3.86 ± 1.35 ^a^	147.77 ± 2.83 ^b^	168.97 ± 0.58 ^c^
Sweetness	Ser (Serine)	4.85 ± 0.02 ^a^	50.17 ± 0.96 ^b^	60.60 ± 0.36 ^c^
	Ala (Alanine)	1.83 ± 0.01 ^a^	60.38 ± 0.18 ^b^	82.29 ± 0.12 ^c^
	Gly (Glycine)	n.d.	49.04 ± 0.47 ^a^	63.05 ± 0.29 ^b^
	Thr (Threonine)	n.d.	33.27 ± 0.20 ^a^	38.33 ± 0.13 ^b^
	Pro (Proline)	n.d.	25.90 ± 0.23 ^a^	37.58 ± 0.51 ^b^
	Total	6.68 ± 0.02 ^a^	218.77 ± 2.03 ^b^	281.85 ± 1.41 ^c^
Bitterness	Arg (Arginine)	n.d.	52.92 ± 1.15 ^a^	58.39 ± 0.18 ^b^
	Tyr (Tyrosine)	n.d.	18.99 ± 0.04 ^a^	24.42 ± 0.00 ^b^
	Val (Valine)	n.d.	25.81 ± 1.32 ^a^	34.83 ± 0.09 ^b^
	Met (Methionine)	9.14 ± 0.22 ^a^	47.35 ± 3.31 ^b^	59.19 ± 2.41 ^c^
	Ile (Isoleucine)	n.d.	17.67 ± 0.05 ^a^	24.06 ± 0.04 ^b^
	Phe (Phenylalanine)	n.d.	25.81 ± 1.19 ^a^	33.06 ± 0.37 ^b^
	Lys (Lysine)	n.d.	36.85 ± 0.41 ^a^	48.53 ± 0.22 ^b^
	Leu (Leucine)	n.d.	32.69 ± 0.48 ^a^	44.15 ± 0.59 ^b^
	His (Histidine)	n.d.	23.68 ± 0.08 ^a^	27.71 ± 0.21 ^b^
	Total	9.14 ± 0.22 ^a^	281.76 ± 8.03 ^b^	354.34 ± 4.10 ^c^
Tasteless	Cys-Cys (Cysteine)	n.d.	4.94 ± 0.25 ^a^	8.81 ± 0.05 ^b^
Total free amino acids		19.68 ± 1.59 ^a^	653.24 ± 13.14 ^b^	813.97 ± 6.14 ^c^
Organic acids				
	Oxalic acid	5.92 ± 0.03 ^a^	105.68 ± 1.93 ^b^	144.35 ± 5.32 ^c^
	Ascorbic acid	n.d.	35.83 ± 0.91 ^a^	47.57 ± 0.87 ^b^
	Lactic acid	n.d.	1112.98 ± 18.36 ^a^	1523.58 ± 33.13 ^b^
	Pyroglutamic acid	n.d.	156.58 ± 3.34 ^a^	179.40 ± 3.13 ^b^
	Citric acid	7.72 ± 0.24 ^a^	3999.96 ± 85.46 ^b^	4692.41 ± 80.44 ^c^
	Succinic acid	n.d.	689.40 ± 13.24 ^a^	927.30 ± 17.77 ^b^
Total organic acids		13.64 ± 0.27 ^a^	6100.43 ± 123.24 ^b^	7514.61 ± 140.66 ^c^
Nucleotides				
	5′-CMP	16.62 ± 2.06 ^a^	80.47 ± 2.27 ^b^	76.27 ± 7.84 ^b^
	5′-GMP	28.75 ± 1.00 ^a^	39.25 ± 0.53 ^b^	42.38 ± 2.82 ^b^
	5′-AMP	10.44 ± 0.97 ^a^	36.76 ± 1.97 ^b^	40.14 ± 8.34 ^b^
	5′-IMP	15.81 ± 0.73 ^a^	20.50 ± 0.16 ^b^	21.52 ± 0.54 ^b^
Total nucleotides		71.62 ± 4.76 ^a^	176.98 ± 4.93 ^b^	180.31 ± 19.54 ^b^

n.d. means not detected. Means in the same row with no common superscript differed significantly (*p* < 0.05).

**Table 3 foods-10-01456-t003:** Mineral contents of different soups.

Elements	Contents (mg/L)		
	Ginger Soup	Chicken Soup	Ginger Chicken Soup
Ca	18.39 ± 0.24 ^a^	88.50 ± 1.22 ^c^	66.83 ± 0.29 ^b^
Cr	0.27 ± 0.00 ^a^	0.84 ± 0.01 ^c^	0.78 ± 0.02 ^b^
Cu	n.d.	0.11 ± 0.00 ^a^	n.d.
Fe	0.94 ± 0.00 ^a^	2.06 ± 0.02 ^b^	n.d.
K	184 ± 2.20 ^a^	1082.50 ± 13.23 ^c^	818.33 ± 2.89 ^b^
Mg	49.45 ± 0.09 ^a^	37.04 ± 0.11 ^b^	28.90 ± 0.09 ^c^
Na	99.63 ± 2.78 ^a^	393.38 ± 6.17 ^c^	290.67 ± 2.02 ^b^

n.d. means not detected. Means in the same row with no common superscript differed significantly (*p* < 0.05).

## Data Availability

The data presented in this study are available on request from the corresponding author.

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
