# Peer review of "Effect of Ginger on Chemical Composition, Physical and Sensory Characteristics of Chicken Soup"

_foods, 2021, doi:10.3390/foods10071456_

Round 1
Reviewer 1 Report
This paper aims to demonstrate the effect of ginger addition on chemical composition and physical and sensory characteristics of chicken soup. The concentration of sugars, amino acids, organic acids, nucleotides and mineral elements as well as physical characteristics including particle size, rheological properties, and color parameters were determined. The flavor profiles of soup were analyzed using electronic tongue.
I would recommend Authors to improve the manuscript according to the following points and make major revision.
The title does not fully correspond to the content of the paper. The paper demonstrates the effect of ginger addition on chemical composition as well as physical and sensory characteristics of chicken soup.
Term “taste substances” is not defined and it is incorrect. According to REGULATION (EC) No 1334/2008 OF THE EUROPEAN PARLIAMENT AND OF THE COUNCIL of 16 December 2008 on flavourings and certain food ingredients with flavouring properties for use in and on foods and amending Council Regulation (EEC) No 1601/91, Regulations (EC) No 2232/96 and (EC) No 110/2008 and Directive 2000/13/EC, the term “flavouring substance” means “a defined chemical substance with flavouring properties”.
I recommend to change “taste substances” to “flavouring substances” throughout the paper.
There are also some mistakes in sensory science vocabulary – please check again the manuscript with an expert in this science. It is incorrect to use terms “taste profiles” (line 22), “taste of three samples” (line 216). These are flavour profiles and flavours of samples. The basic tastes are sweet, bitter, sour, salty and umami.
Abstract could be rewritten to emphasize that the effect of ginger addition on chemical composition and physical and sensory characteristics of chicken soup was studied. Also the aim (lines 63-68) should be rewritten.
Introduction has been presented with only a few number of relevant references. Introduction could be improved.
Lines 13 and 16 – “With the ginger adding” is repeated. Writing style should be improved.
Lines 71-73 - Are these values declared on the packaging? It is not clear.
Line 134 – is “*” should be “a*”
Line 261 – The title of section 3.2 should be corrected
Line 321 – is “6 organic acids” should be “Six organic acids”
Table 2 - There is no explanation of compounds abbreviations below the table
Section 3.10 - The description of the correlation analysis is unclear. All amino acids and all nucleotides were analyzed and only one organic acid (succinic acid). Why the authors excluded the others organic acids, especially citric acid occurring in samples in significant amounts? Why the results of sensory tongue analysis were omitted in the correlation analysis? These results could be of particular interest. Some compounds were analyzed separately (individual nucleotides) and others as a group of compounds (amino acids). What does “color” mean? L*, a*, or b*? This issue requires clarification. The analysis should be repeated taking into account all the parameters investigated.
Fig. 2 is illegible.
Conclusions should be concise (1-2 paragraphs) focusing on the significant, novel findings of the study.
Author Response
Responses to Editor and referees’ comments
Academic Editor:
Please clarify one aspect of the methods prior to peer review of your manuscript. It is indicated that the analytical tests were repeated three times (lines 230 to 231). However the number of batches of chicken soup that were produced and evaluated is not presented. Based on the description on lines 95-101, as written, only one chicken was used for a single batch of soup. Please update your methods to indicate clearly the number of batches of soup that were evaluated in this study.
Response: Thank you for your advice. The information was supplemented and marked in red in the manuscript. The method was uploaded as described in Line 103-113.
“Chicken soup: According to the method of Fan et al., (2020a). All Sanhuang chickens were ca. 1000g in weight and were stored at −20 °C in polyethylene bags until use. The head, claw, neck, and the visible fat were removed according to the naked eye observation. Then the rest part was blanched with boiling water for 1 min, and stewed with 2 times weight of water in a nutrient soup model with the stewing temperature of 100 °C for 3 h. After the chicken soup was cooled to room temperature, it was filtered with a No. 4 qualitative filter paper (Ge Biotechnology (Hangzhou) Co., Ltd.), centrifuged at 10000 r/min for 10 min to remove impurities, and packed into the tube for standby. For each batch of chicken, three replicates were performed, while every two samples of the soups of the same batch of chicken were combined, resulting in three samples of soups for the subsequent analysis.”
- Fan, M., Xiao, Q., Xie, J., Cheng, J., Sun, B., Du, W., Wang, Y., Wang, T. Aroma compounds in chicken broths of Beijing Youji and commercial broilers. J. Agric. Food Chem. 2018, 66, 10242–10251. (Line478-479)
Reviewer: 1
Comment 1: The title does not fully correspond to the content of the paper. The paper demonstrates the effect of ginger addition on chemical composition as well as physical and sensory characteristics of chicken soup.
Response: Thank you for your advice. We have revised the title according to the content.
The title was revised to “Effect of ginger on chemical composition, physical and sensory characteristics of chicken soup”. They are marked in red in the manuscript. (Line2-3)
Comment 2: Term “taste substances” is not defined and it is incorrect. According to REGULATION (EC) No 1334/2008 OF THE EUROPEAN PARLIAMENT AND OF THE COUNCIL of 16 December 2008 on flavourings and certain food ingredients with flavouring properties for use in and on foods and amending Council Regulation (EEC) No 1601/91, Regulations (EC) No 2232/96 and (EC) No 110/2008 and Directive 2000/13/EC, the term “flavouring substance” means “a defined chemical substance with flavouring properties”.I recommend to change “taste substances” to “flavouring substances” throughout the paper.
Response: Thank you for your advice. The ‘taste substances’ was revised to ‘taste components’throughout the paper. According to your advice and related literature, the term “flavouring substance” means “a defined chemical substance with flavouring properties”, it contains not only taste components, but also aroma components. This study of this paper only analyzed the taste compounds in soups, so “taste components” was more accurate. They are marked in red throughout the paper. (Line11,22,24,258)
Comment 3: There are also some mistakes in sensory science vocabulary – please check again the manuscript with an expert in this science. It is incorrect to use terms “taste profiles” (line 22), “taste of three samples” (line 216). These are flavour profiles and flavours of samples. The basic tastes are sweet, bitter, sour, salty and umami.
Response: Thank you very much. We have checked the manuscript with an expert in the science. The flavour profiles include ‘taste profiles’ and ‘aroma profiles’. The purpose of this paper is to study the taste compounds in soups, so we chose the words‘taste profiles’ (Line 22). ‘taste of three samples’was retained. (Line 228-229).
Comment 4: Abstract could be rewritten to emphasize that the effect of ginger addition on chemical composition and physical and sensory characteristics of chicken soup was studied. Also the aim (lines 63-68) should be rewritten.
Response: Thank you very much. The aim of this study has been rewritten.
They are marked in red. (Line 66--72)
The aim was revised to
“The aims of this work were to investigate the effect of ginger on chemical composition and physical and sensory characteristics of chicken soup. By comparing the color, particle size, potential, viscosity, total sugar, amino acids, organic acids, nucleotides and mineral elements of three samples, ginger soup, chicken soup and ginger chicken soup, the influence of ginger on the sensory characteristics of chicken soup could be fully determined. The relationship between chemical composition and physical and sensory quality was also analyzed.”
Comment 5: Introduction has been presented with only a few number of relevant references. Introduction could be improved.
Response: Thank you for your advice. The introduction has been improved. The two relevant references have been supplemented (Line 455, Line 468-469). They are marked in red in the manuscript. (Line 34-35, Line 54-56).
“Chicken is popular for its delicious taste, low fat, high protein and rich in various essential fatty acids. Compared with beef, pork and mutton, chicken is more delicate and richer in nutrients and flavor components. At present, the main cooking methods of chicken include steaming, stewing and boiling. Among them, stewing is more widely used for the soup product. The chicken soup is regarded as a good tonic with therapeutic function [1]. Micro nano colloids in soup have important biological activi-ties, which can change the absorption and utilization behavior of nutrients and bioac-tive compounds [2]. The colloidal characteristics of those nano-particles were preliminarily investigated. It provide a novel perspective to understanding the complicated of soups [3]. Boiling soup can release the nutritional components and flavor substances such as high-quality protein, creatinine, functional peptides, flavor peptides, flavor nucleotides and free amino acids in chicken, making chicken soup delicious and easy to digest and absorb [4,5].
The flavor is often the key factor affecting consumers’ acceptance and preference of food, particularly for soups. Therefore, several studies published in the last two decades have focused on the identification and characterization of the flavor-imparting components of soups. Zhan et al. (2020) analyzed the relationship between non-volatile flavor compounds in chicken soup and sensory evaluation, and they found that aspar-tic acid, glutamic acid, glycine, alanine and proline strongly affected the umami taste in chicken soup [6]. Kong et al. (2017) isolated, purified and identified flavor compounds in chicken soup and chicken enzymatic hydrolysates. They found that the content of flavor compounds in the enzymatic hydrolysate was higher than that in chicken soup [7]. In addition to flavor compounds, sensory characteristic such as color, zeta potential, viscosity could also be used to measure the overall taste of soup. Zou, Xu, Zou, and Yang (2021) compared the differences in color, particle size, zeta potential, viscosity, flavor, mineral, purine and total triglyceride content of soup made by different pro-cessing methods to understand the effect of dietary soup on health, that is, the flavor characteristics of cooked soup were obviously better than that of steamed soup, which proved the importance of cooking technology on soup [8]. Some studies have shown that the release of nutrients in soup is not only attributed to the occurrence of Maillard reaction, but also related to the nanostructured components in soup [9].
- Ke, L., Zhou, J., Lu, W., Gao, G., Rao, P. The power of soups: super-hero or team-work? Trends Food Sci Tech. 2011, 22, 492-497.
- Augustin, M. A., Sanguansri, L., Bode, O. Maillard reaction products as encapsulants for fish oil powders. J Food Sci, 2006, 71, 25-32.
Comment 6: Lines 13 and 16 – “With the ginger adding” is repeated. Writing style should be improved.
Response: Thank you for your advice. The “With the ginger adding” was revised to “Meanwhile”. (Line16)
Comment 7: Lines 71-73 - Are these values declared on the packaging? It is not clear.
Response: These values were not declared on the packaging. These values in Sanhuang chicken were determined according to the national standard method. The ash, crude protein, crude fat, moisture and total sugar were according to GB 5009.3-2016 (the first method), GB 5009.5-2016 (the first method), GB / T 5009.6-2016 (the second method), GB 5009.3-2016 and GB / T 9695.31-2008 (the second method) respectively. (Line77-81)
Comment 8: Line 134 – is “*” should be “a*”
Response: Thank you for your advice. “*” was revised to “a*”. They are marked in red. (Line146)
Comment 9: Line 261 – The title of section 3.2 should be corrected
Response: Thank you for your advice. The title of section 3.2 had been corrected to “Viscosity analysis”. They are marked in red. (Line273)
Comment 10: Line 321 – is “6 organic acids” should be “Six organic acids”
Response: Thank you for your advice. “6 organic acids” was revised to “Six organic acids”. They are marked in red. (Line333)
Comment 11: Table 2 - There is no explanation of compounds abbreviations below the table
Response: Thank you for your advice. The explanations of compounds abbreviations had been supplemented in Table 2. They are marked in red. (Line324)
|
Compounds |
Contents (μg/mL) |
|||
|
Ginger soup |
Chicken soup |
Ginger chicken soup |
||
|
Free amino acids |
||||
|
Umami |
Aspartic acid |
3.86±1.35a |
34.88±2.47b |
46.23±0.23c |
|
Glutamic acid |
nd |
112.90±0.36a |
122.74±0.35b |
|
|
Total |
3.86±1.35a |
147.77±2.83b |
168.97±0.58c |
|
|
Sweetness |
Serine |
4.85±0.02a |
50.17±0.96b |
60.60±0.36c |
|
Alanine |
1.83±0.01a |
60.38±0.18b |
82.29±0.12c |
|
|
Glycine |
nd |
49.04±0.47a |
63.05±0.29b |
|
|
Threonine |
nd |
33.27±0.20a |
38.33±0.13b |
|
|
Proline |
nd |
25.90±0.23a |
37.58±0.51b |
|
|
Total |
6.68±0.02a |
218.77±2.03b |
281.85±1.41c |
|
|
Bitterness |
Arginine |
nd |
52.92±1.15a |
58.39±0.18b |
|
Tyrosine |
nd |
18.99±0.04a |
24.42±0.00b |
|
|
Valine |
nd |
25.81±1.32a |
34.83±0.09b |
|
|
Methionine |
9.14±0.22a |
47.35±3.31b |
59.19±2.41c |
|
|
Isoleucine |
nd |
17.67±0.05a |
24.06±0.04b |
|
|
Phenylalanine |
nd |
25.81±1.19a |
33.06±0.37b |
|
|
Lysine |
nd |
36.85±0.41a |
48.53±0.22b |
|
|
Leucine |
nd |
32.69±0.48a |
44.15±0.59b |
|
|
Histidine |
nd |
23.68±0.08a |
27.71±0.21b |
|
|
Total |
9.14±0.22a |
281.76±8.03b |
354.34±4.10c |
|
|
Tasteless |
Cysteine |
nd |
4.94±0.25a |
8.81±0.05b |
|
Total free amino acids |
19.68±1.59a |
653.24±13.14b |
813.97±6.14c |
|
|
Organic acids |
||||
|
Oxalic acid |
5.92±0.03a |
105.68±1.93b |
144.35±5.32c |
|
|
Ascorbic acid |
nd |
35.83±0.91a |
47.57±0.87b |
|
|
Lactic acid |
nd |
1112.98±18.36a |
1523.58±33.13b |
|
|
Pyroglutamic acid |
nd |
156.58±3.34a |
179.40±3.13b |
|
|
Citric acid |
7.72±0.24a |
3999.96±85.46b |
4692.41±80.44c |
|
|
Succinic acid |
nd |
689.40±13.24a |
927.30±17.77b |
|
|
Total organic acids |
13.64±0.27a |
6100.43±123.24b |
7514.61±140.66c |
|
|
Nucleotides |
||||
|
5´-CMP |
16.62±2.06a |
80.47±2.27b |
76.27±7.84b |
|
|
5´-GMP |
28.75±1.00a |
39.25±0.53b |
42.38±2.82b |
|
|
5´-AMP |
10.44±0.97a |
36.76±1.97b |
40.14±8.34b |
|
|
5´-IMP |
15.81±0.73a |
20.50±0.16b |
21.52±0.54b |
|
|
Total nucleotides |
71.62±4.76a |
176.98±4.93b |
180.31±19.54b |
|
Comment 12: Section 3.10 - The description of the correlation analysis is unclear. All amino acids and all nucleotides were analyzed and only one organic acid (succinic acid). Why the authors excluded the others organic acids, especially citric acid occurring in samples in significant amounts? Why the results of sensory tongue analysis were omitted in the correlation analysis? These results could be of particular interest. Some compounds were analyzed separately (individual nucleotides) and others as a group of compounds (amino acids). What does “color” mean? L*, a*, or b*? This issue requires clarification. The analysis should be repeated taking into account all the parameters investigated.
Response: Thank you for your advice. The other organic acids including citric acids had been supplemented. The sensory tongue analysis was also analyzed, we other amino acids as individual has been supplemented. They marked in red. (Line416-427). The discussion was as follows:
“The correlation analysis was carried out among sensory evaluation and ECU, total sugar, amino acids, organic acid, nucleotides and physical properties (zeta potential, particle size, color, viscosity and shear stress). The taste of chicken soup could be characterized by different indicators. Chicken soup was a mixed system, so its sensory properties would be related to the material composition of the system and then affected the different tastes. As shown in Figure 2, umami had positive correlation with EUC, 5'-IMP, 5'-AMP, organic acids, amino acids, zeta potential and viscosity. Sweetness had negative correlation with organic acids, EUC, nucleotides and amino acids. Saltiness had significantly positive correlation with Asp, Pro, zeta potential and viscosity. EUC values had significantly positive correlation with Asp, Glu, Gly, Pro, Arg, His, 5'-IMP, citric acids, pyroglutamic acid and zeta potential. It also showed that there was a correlation between sensory evaluation and components”.
The color was represented by L*, a*, and b* in three samples. where L * value represents the brightness of the sample (L * = 0 is black, L * = 100 is white), * represents the red and green degree of the sample, b * represents the yellow and blue degree of the sample.
Comment 13: Fig. 2 is illegible.
Response: Thank you for your advice. The legibility of Fig.2 has been adjustied in manuscript.
Comment 14: Conclusions should be concise (1-2 paragraphs) focusing on the significant, novel findings of the study.
Response: Thank you very much. The conclusion has been rewritten.
They are marked in red. (Line 435-440)
The conclusion was revised to
“In summary, the chemical component and physical of chicken soup are related to sensory characteristics. Compared with chicken soup, ginger chicken soup had small particle size, higher viscosity, higher total sugar, lower stability and richer taste. With the ginger adding, the content of total sugar, umami free amino acids and sweet amino acids in chicken soup increased significantly, making chicken soup sweeter and richer. These results proved the importance of ginger to the quality of the chicken soup.”

Reviewer 2 Report
Major comments
- Preparation of soup
You should show the results of ginger's effects, with the experiment of the gradual addition of ginger. Do you have preliminary data? How did you decided the stewing temperature 100℃ for 3 h ? Why was it 2 g ginger (100 g meat) ?
- Electronic tongue analysis
Since you had took the last three response values, you should check the significant difference (p < 0.05) among three different soups
A description of line 396-398 about electronic tongue analysis seemed incorrect.
There were no difference between chicken soup and ginger chicken soup, weren’t there?
It should be described appropriately and shown in the figure.(Fig. S1.→Fig. 1)
In addition, a description of the abstract (line21-22) is inappropriate. It is not clearly stated whether it was influenced by ginger.
- The mechanisms of ginger
You should consider what kind of components contained in ginger affected the amino acids etc.
- Correlation analysis
The number of samples is too small for correlation analysis, and correlation coefficient is unreliable. We recommend that you delete the results of the correlation analysis from this paper. It makes sense if it was associated with sensory evaluation.
Minor comments
HPLC, ICP, Electric Tongue → Please describe the company name of the device and its address.
Author Response
Responses to Editor and referees’ comments
Academic Editor:
Please clarify one aspect of the methods prior to peer review of your manuscript. It is indicated that the analytical tests were repeated three times (lines 230 to 231). However the number of batches of chicken soup that were produced and evaluated is not presented. Based on the description on lines 95-101, as written, only one chicken was used for a single batch of soup. Please update your methods to indicate clearly the number of batches of soup that were evaluated in this study.
Response: Thank you for your advice. The information was supplemented and marked in red in the manuscript. (Line 103-113) The method was uploaded as described in Line 103-113.
“Chicken soup: According to the method of Fan et al., (2020a). All Sanhuang chickens were ca. 1000g in weight and were stored at −20 °C in polyethylene bags until use. The head, claw, neck, and the visible fat were removed according to the naked eye observation. Then the rest part was blanched with boiling water for 1 min, and stewed with 2 times weight of water in a nutrient soup model with the stewing temperature of 100 °C for 3 h. After the chicken soup was cooled to room temperature, it was filtered with a No. 4 qualitative filter paper (Ge Biotechnology (Hangzhou) Co., Ltd.), centrifuged at 10000 r/min for 10 min to remove impurities, and packed into the tube for standby. For each batch of chicken, three replicates were performed, while every two samples of the soups of the same batch of chicken were combined, resulting in three samples of soups for the subsequent analysis.”
- Fan, M., Xiao, Q., Xie, J., Cheng, J., Sun, B., Du, W., Wang, Y., Wang, T. Aroma compounds in chicken broths of Beijing Youji and commercial broilers. J. Agric. Food Chem. 2018, 66, 10242–10251. (Line478-479)
Reviewer 2:
Comment 1: Preparation of soup
You should show the results of ginger's effects, with the experiment of the gradual addition of ginger. Do you have preliminary data? How did you decided the stewing temperature 100℃ for 3 h ? Why was it 2 g ginger (100 g meat) ?
Response: Thank you for your advice. We did the pre-experiment. The processing method of stewed ginger chicken soup was optimized by single factor and orthogonal test. Taking the sensory evaluation results as the reference index, three factors of material liquid ratio, stewing time and ginger content were investigated. We finally determined that the optimal stewing time was 3.0 h, the ratio of material to liquid was 1:2 (m:V), and the amount of ginger was 2 g (100 g meat).
Table 1 Results of orthogonal experiment on technology optimization of soup
|
Number |
Time(h) |
Material liquid ratio |
Ginger content(g) |
Sensory evaluation score |
|
1 |
3.0 |
1:1.5 |
1 |
6.07±0.18 |
|
2 |
3.0 |
1:2 |
2 |
7.21±0.26 |
|
3 |
3.0 |
1:2.5 |
3 |
4.43±0.53 |
|
4 |
4.0 |
1:1.5 |
2 |
6.36±0.47 |
|
5 |
4.0 |
1:2 |
3 |
4.71±0.48 |
|
6 |
4.0 |
1:2.5 |
1 |
5.43±0.53 |
|
7 |
5.0 |
1:1.5 |
3 |
3.93±0.18 |
|
8 |
5.0 |
1:2 |
1 |
4.43±0.53 |
|
9 |
5.0 |
1:2.5 |
2 |
4.86±0.37 |
|
K1 |
5.90 |
4.67 |
5.31 |
|
|
K2 |
5.50 |
4.92 |
6.14 |
|
|
K3 |
4.40 |
4.90 |
4.36 |
|
|
R |
1.50 |
0.25 |
1.78 |
|
|
Results |
3.00 |
1:2 |
2.00 |
7.21 |
Comment 2: Electronic tongue analysis
Since you had took the last three response values, you should check the significant difference (p < 0.05) among three different soups.
Response: Thank you for your advice. The significant difference (p < 0.05) among three different soups has been added. They are marked in Figure. S1 in manuscript.
Fig. S1. Radar chart of electronic tongue data with different soups. represent ginger soup, represent chicken soup, represent ginger chicken soup.
Comment 3: A description of line 396-398 about electronic tongue analysis seemed incorrect.
Response: Thank you for your advice. The description of after-B was the aftertaste of umami taste, which also showed a good taste to endowed chicken soup. So it is reasonable. (Line 410-412).
Comment 4: There were no difference between chicken soup and ginger chicken soup, weren’t there? It should be described appropriately and shown in the figure.(Fig. S1.→Fig. 1)
Response: Thank you for your advice. We have done significant analysis in Figure. S1., and there are significant differences between them. The described information has been supplemented in manuscript. “Among them, there were significant difference in bitterness, astringency, aftertaste-B, umami, richness and saltiness between chicken soup and ginger chicken soup”. (Line 405-407). And the significance has been marked in Figure. S1. in manuscript.
Comment 5: In addition, a description of the abstract (line21-22) is inappropriate. It is not clearly stated whether it was influenced by ginger.
Response: Thank you for your advice. According to significant analysis in Figure. S1., the taste components and sensory quality of chicken soup were obviously affected by adding ginger. So the description of the abstract (Line21-22) was retained.
Comment 6: The mechanisms of ginger
You should consider what kind of components contained in ginger affected the amino acids etc.
Response: Thank you for your advice. According to the reference, in the process of boiling soup, the rich content of proteins, and saccharides in plant materials facilitates the reaction. So the polysaccharides and rude proteins in ginger would affect amino acids etc.
The reference has been supplemented to manuscript. (Line 455)
- Ke, L., Zhou, J., Lu, W., Gao, G., Rao, P. The power of soups: super-hero or team-work? Trends Food Sci Tech. 2011, 22, 492-497.
Comment 7: Correlation analysis
The number of samples is too small for correlation analysis, and correlation coefficient is unreliable. We recommend that you delete the results of the correlation analysis from this paper. It makes sense if it was associated with sensory evaluation.
Response: Thank you for your advice. Correlation analysis refers to the analysis of two or more variable factors with correlation to measure the closeness of the two variable factors. The existing data through correlation analysis could be concluded that there was certain correlation between sensory evaluation and each factor. The sensory tongue analysis has been supplemented. They marked in red. (Line416-427). The discussion was as follows:
“The correlation analysis was carried out among sensory evaluation and ECU, total sugar, amino acids, organic acid, nucleotides and physical properties (zeta potential, particle size, color, viscosity and shear stress). The taste of chicken soup could be characterized by different indicators. Chicken soup was a mixed system, so its sensory properties would be related to the material composition of the system and then affected the different tastes. As shown in Figure 2, umami had positive correlation with EUC, 5'-IMP, 5'-AMP, organic acids, amino acids, zeta potential and viscosity. Sweetness had negative correlation with organic acids, EUC, nucleotides and amino acids. Saltiness had significantly positive correlation with Asp, Pro, zeta potential and viscosity. EUC values had significantly positive correlation with Asp, Glu, Gly, Pro, Arg, His, 5'-IMP, citric acids, pyroglutamic acid and zeta potential. It also showed that there was a correlation between sensory evaluation and components”.
Comment 8: Minor comments
HPLC, ICP, Electric Tongue→Please describe the company name of the device and its address.
Response: Thank you for your advice. The company name of the device and its address have been supplemented. HPLC (Agilent Technology Co., Ltd, China) (Line168), HPLC (Thermo Scientific, U.S.A.) (Line184), ICP (Shimadzu, Japan) (Line221), Electric Tongue (Intelligent Sensor Technology, Inc., Kanagawa, Japan). They are marked in red. (Line 227-228)
